# Stakeholders’ Understanding of European Medicine Agency’s COVID-19 Vaccine Information Materials in EU and Regional Contexts

**DOI:** 10.3390/vaccines11101616

**Published:** 2023-10-19

**Authors:** Indiana Castro, Marie Van Tricht, Nicole Bonaccorso, Martina Sciortino, Juan Garcia Burgos, Claudio Costantino, Rosa Gonzalez-Quevedo

**Affiliations:** 1Public and Stakeholder Engagement Department, European Medicines Agency, Domenico Scarlattilaan 6, 1083 HS Amsterdam, The Netherlands; 2Department of Health Promotion Sciences, Maternal and Infant Care, Internal Medicine and Medical Specialties (PROMISE) “G. D’Alessandro”, University of Palermo, 90127 Palermo, Italy; martina.sciortino@unipa.it (M.S.); claudio.costantino01@unipa.it (C.C.)

**Keywords:** vaccination, COVID-19, EMA, misinformation, hesitancy

## Abstract

The COVID-19 pandemic posed challenges to communicating accurate information about vaccines because of the spread of misinformation. The European Medicines Agency (EMA) tried to reassure the public by communicating early on about the development and approval of COVID-19 vaccines. The EMA surveyed patients/consumers, healthcare professional organizations, and individual stakeholders, both at the EU level and in an Italian regional context. The objectives of the study were to see if the EMA’s core information materials were informative and well-understood and which communication channels were preferred by the public. The main findings showed that individual patients/consumers generally prefer to obtain information about COVID-19 vaccines from the internet or mass media, while organizations and individual healthcare professionals prefer to obtain information from national and international health authorities. Both at EU and local levels, participants had a good understanding of the key messages from regulators and found the materials useful and relevant. However, some improvements were recommended to the visual, text, and dissemination formats, including publishing more information on safety and using a more public-friendly language. Also, it was recommended to maintain the EMA’s approach of using media, stakeholder engagement, and web-based formats to communicate about COVID-19 vaccines. In conclusion, user-testing of proactive communication materials aimed to prebunk misinformation during a public health crisis helps to ensure that users understand the development and safety of novel vaccine technologies. This information can then be used as a basis for further evidence-based communication activities by regulators and public health bodies in an emergency context.

## 1. Introduction

Severe acute respiratory syndrome coronavirus 2 (SARS-CoV-2) is the causative agent of the current COVID-19 global pandemic, proclaimed a public health emergency of global concern by the World Health Organization (WHO) between March 2020 and May 2023 [1].

As of 21 September 2023, there have been 770,778,396 confirmed cases of COVID-19, including 6,958,499 confirmed deaths caused by SARS-CoV-2 since the beginning of the pandemic [2]. In addition, an estimated over 26 million excess deaths linked to the pandemic have been reported, apart from the emerging long-term sequelae, referred to as post-COVID-19 condition (PCC), long COVID, or post acute sequelae of COVID-19 (PASC) [3,4,5,6].

As the COVID-19 pandemic emerged and it became clear that vaccines against SARS-CoV-2 would be essential in managing severe outcomes and alleviating pressure on healthcare systems [7], public interest was raised in potential vaccine developments and technologies that would enable the fast delivery of vaccines. This was accompanied by concerns in media and social media on the speed at which these vaccines would be developed and approved. Already at these early stages, there was speculation on potential opposition to vaccines developed at pandemic speed [8,9] and the challenge of achieving a high vaccination uptake was highlighted. This was particularly acute as many parts of the world, including Europe, were already reeling from the effects of vaccine hesitancy, which had prompted the European Commission (EC) and European Union (EU) member states to launch several initiatives in 2018 to improve confidence in vaccines for routine immunizations [10]. Early in the pandemic, academics and public health experts alike highlighted the need for proactive communication to anticipate (“prebunk”) potential misinformation or deliberate attempts to disinform the public [11].

With the approval of the first vaccine in the EU against COVID-19 via the European Medicines Agency (EMA, also referred to as “The Agency”) [12] and after the start of vaccination campaigns by EU national health regulatory authorities, a significant decrease in severe disease and deaths has been observed thanks to vaccination. In particular, a greater reduction was observed in the countries where vaccination reached high coverage rates, while in EU countries that lagged behind, hesitancy remained an obstacle to vaccine uptake and control of the pandemic [13]. Despite vaccines’ protective effect against severe disease and death, concerns are regularly exploited by misinformation/disinformation sources on other aspects such as very rare and serious adverse events.

To date, several COVID-19 vaccines are authorized in the EU, four of them using new technologies for vaccine development [14]. Worldwide, as of 4 July 2023, almost 13.5 billion doses of the vaccines have been administered [2]. As of 11 July 2023, 330,874,923 people in the EU/European Economic Area countries have received a primary vaccination course, which accounts for approximately 73% of the total population, and approximately 65% of adults (18+) have received at least one booster/additional dose [15].

However, despite differences across EU member states, a large part of the population remains unvaccinated, often due to concerns aggravated by disinformation about vaccines, including concerns about the safety of new vaccine platforms and expedited vaccine development [16,17].

In the context of a reality characterized by uncertainty about how the COVID-19 situation will evolve in light of new variants, the emergence of waning immunity and a large unvaccinated population, mis/disinformation remains a challenge [17,18,19]. The Agency, as the regulatory authority in the EU responsible for evaluating COVID-19 vaccines and their safety monitoring, has communicated extensively on their development and approval in short timeframes to reassure the population that all regulatory standards are complied with and that patient safety is prioritized [19]. In the present study, the initial information materials prepared by the Agency have been user-tested to check whether they were informative and understood by EU patients, consumer organizations (PCOs), and healthcare professional organizations (HCPOs) during the initial introduction of COVID-19 vaccines. The work of national medicine regulatory bodies is critical for public outreach at national and local levels. Therefore, to complement the study with the national context, similar stakeholder groups were surveyed in a local hospital setting at the University Hospital of Palermo, Italy. The findings in this study highlight the need for evidence-based communication approaches during a public health crisis and in a preparedness setting.

## 2. Materials and Methods

### 2.1. Materials

This research aimed to evaluate two documents prepared by the EMA addressing common questions about COVID-19 vaccines: “Tier 1” or “COVID-19 vaccines: basic facts” and “Tier 2” or “COVID-19 vaccines in the EU: development, evaluation, approval and monitoring” (Appendix A).

The document “COVID-19 vaccines: key facts” covers the initial high-level information about COVID-19 vaccines, including why they are needed and basic information on the approval and effectiveness of COVID-19 vaccines. 

The second document, “COVID-19 vaccines: development, evaluation, approval and monitoring”, contains more detailed information on the lifecycle of COVID-19 vaccines, the accelerated timelines of COVID-19 vaccines compared to regular approval, and the process for post-marketing safety monitoring. 

These two documents served as core information used by the EMA to communicate with the public when COVID-19 vaccines were first approved, including through the EMA’s website [20,21].

### 2.2. Sample

The EU survey was sent by targeted email to individual patients, consumers, and healthcare professionals (HCPs) as well as EU organizations representing these groups. In addition, it was also sent by targeted email to a pool of volunteers with confirmed COVID-19 cases who had offered to assist the Agency in COVID-19-related work relevant to these stakeholder groups. All survey recipients were part of the regular and voluntary group of stakeholders registered in the EMA stakeholders’ database [22]. At the time of this analysis, the survey was sent to 35 healthcare professional organizations, 37 patients and consumer organizations, and 26 individual patients and consumers. 

The stakeholders representing the regional context (Sicily) in an EU member state (Italy) are a corresponding number of individual patients, consumers, and HCPs as well as national and regional organizations representing Italian stakeholders. Sicily, the identified regional context, is a southern Italian region with approximately 5 million inhabitants and ranks fourth in terms of demographic density in Italy. In particular, the province of Palermo, accounting for 1,214,291 inhabitants residing in 82 municipalities, is the most populous in the region [23]. Since the beginning of the COVID-19 vaccination campaign in Italy (28 December 2020), the vaccination center of the University Hospital of Palermo has been one of the vaccination hubs for HCPs and the general population. To date, it has vaccinated more than 85,000 people at the University Hospital of Palermo with at least one dose of COVID-19 vaccines authorized by the EMA. In the Palermo province, 91.9% of the general population received at least one dose (1,115,789 million inhabitants), and over 2.5 million combined first, second, or booster doses of COVID-19 vaccines were administered since the beginning of the campaign. Overall, 7.6% of the general population of the Palermo province was vaccinated at the University Hospital of Palermo. In Italy, more than 145 million doses were administered to the general population with overall vaccination coverage of 91.76% for the first dose, 90.25% for completion of the primary cycle, 84.88% for the first booster, and 16.88% for the second booster doses, respectively [24]. 

### 2.3. Survey

The survey was created using EUSurvey, the EC’s online survey management system [25]. The survey link and both Tier 1 and Tier 2 documents in PDF format (Appendix A) were sent via email to the EMA stakeholders’ database and pool of volunteers with confirmed COVID-19 cases (survey questions can be found in Appendix A). For the survey conducted in the regional context, the questionnaire was also translated into Italian (if requested by the respondents) and administered in online or paper form (Appendix A).

The survey was divided into two sections: the first one comprised 12 questions on demographic data, participants’ preferred information sources, and the Tier 1 document “COVID-19 vaccines: basic facts”. The second part of the survey comprised 11 questions on the Tier 2 document “COVID-19 vaccines: development, evaluation, approval and monitoring”. The most relevant questions for this research in both sections aimed to (i) evaluate the understanding of the reader and (ii) obtain overall feedback on the text and some visuals in the documents. The email instructed the participants to read the documents before answering the survey and that the second part of the survey was optional for individual patients, consumers, and volunteers with confirmed COVID-19, as there was more information that contained more scientific and regulatory details than Tier 1. The EU survey remained open between 24 November and 11 December 2020 and between 24 February and 5 March 2021 for the Sicilian regional survey. Quantitative data were collected through multiple choice questions and 7-point Likert scales: (1) not at all, (2) disagree, (3) somewhat disagree, (4) neutral, (5) somewhat agree, (6) agree, (7) absolutely. Qualitative information was collected through free-text boxes. 

### 2.4. Data Analysis

Both quantitative and qualitative analyses were undertaken for different questions of the survey. Quantitative data were analyzed anonymously in Excel using descriptive statistics. No statistical testing was performed given the small number of respondents. Qualitative data were collected through open-ended questions. Researchers (I.C. and R.G.-Q.) analyzed the responses and summarized common themes regarding suggestions and comments for both documents. Validation of the findings was carried out independently by M.V.T. and J.G.B. from EMA and C.C., N.B., and M.S. from the University of Palermo collaborating group. To ensure that there was no potential sample size bias for the different stakeholder groups, responses from the survey were weighted neutrally, regardless of the size of the stakeholder.

### 2.5. Personal Data Protection Statement

The survey instructions stated that no internet protocol addresses would be collected, the data would only be analyzed in an aggregated manner, and the only personal information required was to specify which stakeholder group they belonged to. No information, including whether these participants were COVID-19 patients, was kept for ana-lysis.

## 3. Results

### 3.1. Survey Participant Profiling

A survey was sent to EU stakeholders to determine if both Tier 1 and Tier 2 documents were informative and well understood and to explore the public’s preferred communication channels. A total of 30 participants completed the EU survey, as shown in Figure 1a. Thirteen participants from the HCPO participated in the survey (around 37% response rate). The next stakeholder group with the most participation was the PCO, with eight respondents (around 22% response rate). One of the respondents ticked the option “other” for the stakeholder group questions (see survey questions on the Appendix A), specifying that they were part of an organization mainly grouping researchers in a specific therapeutic area. Researchers categorized this respondent as a PCO because their organization provided patient representation to EMA committees. The response rate by individuals was around 35%.

The survey for regional Italian stakeholders was also completed by 30 participants (response rate 88%), as shown in Figure 1b. From the HCPO, eight representatives participated in the survey (around 98%), while twelve were individual HCPs (around 88%). The next stakeholder group with the most participation was individual patients, with seven respondents (around 86%). Finally, three of the respondents (around 80%) were representatives of patient/consumer organizations. 

In both settings, 23 out of 30 participants completed both sections of the survey, Tier 1 and Tier 2 (Figure 2).

### 3.2. Preferred Information Sources for Patients, Consumers, Healthcare Professionals, and Organizations at EU and Regional Levels

Participants in both the EU and Italian surveys where asked which sources they would consult to find information about COVID-19 vaccines, and similar trends were observed in both settings (Figure 3). 

#### 3.2.1. Individual Patients/Consumers and PCOs (Figure 3a)

In the case of both the EU level and the regional level, for individual patients and consumers, their most consulted sources were internet search engines (respectively 50% and around 57%), newspapers (respectively 50% and around 43%), and their family doctor (respectively 50% and around 43%), followed by scientific journals (around 33%), with others (approximately 33%) not knowing where to look for information. 

EU-level PCOs on the other hand preferred to consult scientific journals (88%), followed by EMA sources (75%), their family doctor, their country’s ministry of health, and the WHO (50%). Regional PCOs indicated a similar preference to consult scientific journals and EMA sources (100%); around 67% of them also used internet search engines, family/colleagues/friends, and newspapers as sources of information. 

#### 3.2.2. Individual HCPs and HCPOs (Figure 3b)

The EU-level individual HCPs’ main preferences included consulting scientific journals, the WHO, and their country’s medicines regulatory agency (around 67%). Only approximately 33% indicated that they consulted the EMA for information. The EU HCPOs preferred to consult the EMA (around 85%), the WHO (around 62%), their country’s medicines regulator agency (around 54%), and the European Centre for Disease Prevention and Control (ECDC) (around 46%), followed by their family doctor and scientific journals (around 38%). 

Both individual HCPs and HCPOs at the regional level mainly consulted the Italian Medicines Agency (Agenzia Italiana del Farmaco, AIFA) (respectively 75% and around 87.5%), the Italian Ministry of Health (respectively 50% and 75%), and scientific journals (respectively approximately 42% and 63%). Similar to the EU context, only around 33% of regional individual HCPs said that they used the EMA as a source of information. On the other hand, around 63% of the HCPOs consulted the EMA for information.

In general, the preferred sources of information among stakeholder groups showed similar trends between EU-level and regional participants. The individual patients and consumers consulted mass media sources more often while the organizations and individual HCPs consulted multiple sources and professional audiences, especially health authorities. These trends are important when considering how to reach a particular stakeholder group.

### 3.3. The Understanding of Key Messages by Stakeholders at EU or Regional Levels 

Participants were asked to read several key messages in Tier 1 and 2 materials to determine if these were clear and understandable. The messages chosen for user testing referred to key concepts that the EMA identified as potentially being able to be misinterpreted or subject to misinformation (Table 1).

Three Tier 1 messages were presented. First of all, a safety message was included to address the misperception that the speedy development of COVID-19 vaccines could entail that corners would be cut with safety testing. Secondly, a message on the approval process was included to address existing concerns that COVID-19 vaccines are not properly studied. This message reiterated that a scientific evaluation of the vaccines is performed by the EMA. Furthermore, an explanation of all elements of the EMA response to ensure the quality, safety, and efficacy of vaccines was provided. 

The three messages of the Tier 2 material focused firstly on the timeline of the safety monitoring of COVID-19 vaccines; secondly on the explanation of a rolling review, a new regulatory process that enables the Agency to assess data as they arise and expedite the process; and lastly on factors that helped the development of COVID-19 vaccines. 

For Tier 1, the results showed that the messages on the safety requirements for COVID-19 vaccines were well understood. However, when asked a more complex question regarding how the EMA ensures that COVID-19 vaccines are safe and effective, the participants scored poorly in both contexts, with only around 27% of EU-level stakeholders and around 35% of regional stakeholders answering correctly. The complexity of the question and the fact that the information needed to provide an answer was spread across the Tier 1 document and therefore not easy to find might explain these results. 

The three messages of Tier 2 were well understood. The answer distribution among stakeholders and between EU-level and regional participants was uniform, suggesting that, overall, the key messages were well understood among the participants.

### 3.4. Participants’ Feedback on Usefulness, Language, and Level of Detail 

EU- and regional-level participants provided feedback on the understanding of the information, the language used, the level of detail provided, and the usefulness and clarity of the information of both Tier 1 and Tier 2. The percentages of people that indicated “absolutely” and “agree” with the survey questions were summarized to obtain the following results. As explained in Figure 2, not all participants responded to the questions related to Tier 2, explaining the difference in the total amount of respondents between Tier 1 and Tier 2.

#### 3.4.1. Feedback on Tier 1

A total of 54% of EU survey participants (16 out of 30) indicated an increase in their understanding of COVID-19 vaccines after reading the document. A total of 60% (18 out of 30) thought the language and vocabulary were easy to understand. In addition, 80% (24 out of 30) were of the opinion that Tier 1 contained enough information to understand the basic facts of the document, as shown in Figure 4a. A total of 80% (24 out of 30) also indicated that they found the information to be both clear and helpful. 

For the regional survey, 67% (20 out of 30) of respondents indicated an increase in their understanding of COVID-19 vaccines after reading the document and 80% (24 out of 30) found the language and vocabulary easy to understand. Furthermore, 73% (22 out of 30) of respondents thought enough information was provided in Tier 1 to understand the key facts of the document and 83% (25 out of 30) found the information clear and helpful. 

These results showed that information in Tier 1 was easy to understand and increased an important proportion of stakeholders’ understanding of COVID-19 vaccines, both at the EU and at regional levels.

#### 3.4.2. Feedback on Tier 2

For Tier 2, 61% (14 out of 23) of the participants in the EU survey declared that their understanding of the COVID-19 vaccines increased after reading the document and 57% (13 out of 23) found the language and vocabulary to be easily understandable. A total of 74% of EU survey respondents (17 out of 23) were of the opinion that enough information was provided. The information was found clear and helpful by 74% of respondents (17 out of 23) (Figure 4b). 

For the regional survey, 83% (19 out of 23) indicated an increase in their understanding of COVID-19 vaccines after reading the document and 65% (15 out of 23) considered the language and vocabulary easy to understand. In addition, 82% (19 out of 23) thought that the document reported enough information to understand the basic facts and 83% (14 out of 23) found that the graphics were clear and helpful. 

These results show that the Tier 2 document also helped with the understanding of COVID-19 vaccines in an important proportion of stakeholders, even considering that this was a longer and more technical document than Tier 1.

When comparing the EU-level and regional responses, the results suggest that, in the context of this study, in general, the regional participants had a stronger positive opinion of both documents in terms of usefulness, clarity, and language.

### 3.5. Participants’ Feedback on Visuals to Convey Regulatory Principles or Processes

Stakeholders have often requested the use of visuals to explain regulatory concepts or processes and the Agency prepared some visual materials for its initial COVID-19 vaccine information. The Tier 2 materials included a number of figures that the Agency has subsequently repurposed in a variety of materials on COVID-19 vaccines (see Appendix A). In Figure 5, when asked about whether visuals and graphic representations were generally considered helpful, 86% (20 out of 23) EU-level and 61% (14 out of 23) regional participants found the graphics helpful in conveying the key concepts.

Participants were asked to review some alternatives for graphics included in Tier 2. Two visuals representing a pooling of resources and the positive benefit–risk balance were user-tested as these were the visuals that generated the most discussion during their preparation. EU-level participants found both pooling of resources visuals clear and easy to understand (65%; 15 out of 23), while the regional participants did not (74% answered “no” or “don’t know”; 17 out of 23) (Table 2). 

When asked which of the two visuals representing the pooling of resources they preferred, the second visual representing an organized pooling of resources was the preferred visual for EU- and regional-level participants (69% and 100%, respectively).

Three visuals representing the concept of a positive benefit–risk balance were user-tested, with 87% (20 out of 23) and 74% (17 out of 23) of EU and regional participants agreeing, respectively, that they were clear and easy to understand. Participants preferred the visual representing a scale weighting benefits versus risks, with 60% (12 out of 20) of EU and 83% (15 out of 18) of regional participants selecting this visual over the other two options.

### 3.6. Participants’ Preferred Statement about COVID-19 Vaccine Safety

The survey also user-tested two key messages that the EMA can use to explain COVID-19 vaccine safety when considering the benefits (Table 3). Both statements explain the same message, but the wording differs. At the EU level, the first safety statement [*Like any medicine, vaccines have benefits and risks, and although highly effective, no vaccine is 100% effective in preventing a disease or 100% safe in all vaccinated people*] was preferred by individual patients and consumers (100%; 2 out of 2) and HCPOs (82%; 9 out of 11). The second safety statement [*Like any medicine, vaccines have benefits and risks, and the conditions of marketing authorization aim at maximizing benefits and minimizing risks*] was preferred by the individual HCPs (100%; 2 out of 2). PCOs preferred either statement (4 out of 8).

At the regional level, the first safety statement was preferred by HCPOs (57%; 4 out of 7). The second safety statement was preferred by individual patients and consumers (60%; 3 out of 5) and individual HCPs (64%; 7 out of 11). 

On the basis of this feedback, and given that PCOs had no preference for either statement, the EMA has decided to use the first safety statement proposal, given that HCPOs preferred it, and healthcare professionals are normally the first point of contact for questions on vaccines and they deal with vaccination-hesitant individuals. 

### 3.7. Feedback on Shareable/Audiovisual Materials, Translation into Official EU Languages, and Missing Information

The survey further explored open feedback from stakeholders on ideas for other types of formats, translations into EU languages, and whether any key information was missing.

To identify other possible formats that could be used to share COVID-19 vaccine information, the survey included a question to see if the participants would be interested in having the Tier 1 information “COVID-19 vaccines: basic facts” in another shareable format besides the webpage (Appendix A, Figure 1). Around 63% of EU-level participants (19 out of 30) and around 67% of regional stakeholders (23 out of 30) responded positively to this. Two Italian respondents elaborated that an updated document should be simpler and with more concise language. PDF documents, videos, and social media were suggested by participants as types of shareable formats they would be interested in. 

Although most of the participants (>70%) stated that they did not have an issue understanding the documents in English (Figure 6), 4% of European-level participants (1 out of 30) and 20% of Italian participants (4 out of 20) indicated that the English language of the materials was an obstacle to understanding the document. 

The survey also asked participants to identify areas where more information was needed, as part of an open question. A recurrent missing item identified by stakeholders was more information on COVID-19 vaccine safety (requested by two EU-level HCPOs, one EU-level individual HCP, and two EU-level PCOs). The materials subject to user-testing from Tier 1 and Tier 2 were non-product-specific and were published before individual vaccines were approved and rolled out in vaccination campaigns; thus, there was no safety information available from real-world data at the time. The EMA has since published regular public safety updates on the approved vaccines on the corporate website, after mass vaccination campaigns had taken place and clinical data had become available [26]. Public safety updates of COVID-19 vaccines are a specific tool that the EMA developed to keep stakeholders informed on safety aspects during the most acute phase of the pandemic and were published as an additional transparency measure to all the information that the EMA routinely publishes on medicines approved via the Agency. They are available under the tab ‘Safety updates’ on the specific vaccine page for the first COVID-19 vaccines approved via the EMA.

### 3.8. Recommendations from the User-Testing Analysis 

The quantitative data showed an overall satisfaction of the participants with both EMA documents (Tier 1 and Tier 2) explaining COVID-19 vaccines. The quantitative and qualitative results from the survey were used to draw recommendations for regulators to improve information materials according to areas of improvement or missing information identified by participants. 

In Table 4, a summary of the actions derived from the survey results and how the EMA has implemented such recommendations is given. As most participants preferred benefit–risk visual A, the “COVID-19 vaccines: development, evaluation, approval and monitoring” webpage and document were updated, replacing the initial figure chosen (Figure C in Table 2) (balance suggested by comparing sizes) by A (balance suggested using a scale). 

Some text and headings were updated to improve understandability in response to the substantially incorrect answers to one comprehension question about “COVID-19 vaccines: fey facts” and to the participants’ calls to use as much lay language as possible. Due to the fact that these materials were created in the context of a public health emergency, it was not possible to conduct a user-testing study before publishing these materials. However, by introducing retrospective improvements, the EMA has been able to address the public’s information needs and formatting preferences captured in this study. 

## 4. Discussion

This research aimed to user-test the EMA’s core information materials on COVID-19 vaccines. More specifically, it tested whether the language was clear, whether enough information was provided, and whether it increased stakeholders’ understanding of how COVID-19 vaccines were developed and approved in a short period without compromising safety, efficacy, and quality standards. We have also user-tested some key graphics and messages used by EMA to communicate on these vaccines to the public.

These materials were proactively prepared before the approval of the first vaccines and were subsequently repurposed by the EMA through different channels and communication tools during the COVID-19 pandemic. As such, the EMA webpages containing the information subject to this study have been regularly visited since their publication, with visit peaks coinciding with the start of the COVID-19 vaccination campaigns in Europe, announcements of COVID-19 vaccine approvals, start of rolling reviews, and evaluation of safety signals (unpublished data).

Subjecting the EMA’s core content to user-testing is relevant not only for selecting messages and figures for webpages but also because this content is repurposed by regulators in response to queries from patients and HCPs, media interviews, and explaining vaccine development and regulatory oversight during public stakeholder meetings held by the EMA. 

These messages and graphics were included in public webpages before the first COVID-19 vaccine was approved in the EU to anticipate mis- and disinformation, as well as to address legitimate stakeholders’ questions and concerns circulating on (social) media with regard to the fast-tracked development, evaluation, and approval of COVID-19 vaccines (18,19). As described by Blastland et al., obtaining a better understanding of the target audience and their concerns is required to be able to prebunk misinformation [11].

When trying to establish how applicable our results can be to the general EU population, or to different stakeholder groups, it is important to examine who responded to the survey. The EU-level participants were predominantly members of patient or healthcare professional organizations, followed by individual patients, consumers, and HCPs. The EMA routinely uses stakeholders registered in its stakeholders’ database for consulting with PCOs or HCPOs different initiatives and regulatory issues and for disseminating information to organizations. The response rates obtained are aligned with responses generally obtained by the EMA with these types of surveys. In particular, the response rates for HCPOs (around 37%) and PCOs (around 22%) were higher than what has been found in other EMA communication research initiatives [26]. 

This proportion differed in the regional context, where most respondents were individual HCPs, followed by HCPOs and individual patients and consumers. This suggests a broad profile of participants; representatives from organizations tend to be more familiar with regulatory processes and language, while individual patients or HCPs tend to be less familiar with them. We can then assume a certain balance in our results.

In general, during the first phases of the European vaccination campaign against COVID-19, individual HCPs and HCPOs played a key role in promoting correct information and counselling on the COVID-19 vaccines available [27,28]. Moreover, they represented, in all EU countries, the first target of vaccination and, consequently, they have become a model for specific subgroups of the general population that were subsequently vaccinated (elderly, fragile, and high-risk groups, etc.) [29,30]. At the same time, the materials published on the EMA’s website were, especially at the regional level, consulted more by HCPs (individuals or organizations) and PCOs. 

The choice to compare the EU with the regional Italian context data was relevant in order to conduct an in-depth evaluation of the readability and reproducibility of the user-testing material in the local context, where there could be participants with a limited knowledge of English, the working language of the EMA [31]. From the results obtained at the local level, for both the individual stakeholders and the PCOs, the language did not represent a considerable limit in the comprehension of the contents of the document. The extensive use of graphics to support the content of the user-tested materials may be a contributing factor to this. The Agency will, however, look longer term into increasing multilingual information materials.

In order to guarantee an appropriate data comparison, the authors tried to administer the survey to similar patterns of respondents at EU and regional levels. Effectively, a similar distribution of representatives and individual HCPs and/or of consumer and patient organizations was observed. Moreover, in both cases, 23 participants out of 30 completed Tier 1 and 2 of the survey. The possibility of translating Tier 1 and 2 content into local EU languages could be considered for documents that are made available for the general public in order to promote a wide dissemination of information, also among people with limited English knowledge or low health literacy levels. The current user-testing of materials was carried out on a population of HCPs (individuals or representatives) and representatives of PCOs that may have a better knowledge of the English language and a better capacity to understand scientific language and regulatory processes (as also reaffirmed by the results obtained). 

Based on the participants who responded to this survey, individual patients and consumers mainly consulted internet search engines, newspapers, and their family doctor as a source of COVID-19 vaccine information. It therefore seems crucial that regulators use these same channels that are easily accessible to the general population in order to effectively counteract the circulation of misinformation. As for doctors themselves, it is also necessary for them to receive complete and clear information from authoritative and unbiased sources since they represent a point of reference for the population, as also shown by the results of the ADVANCE project [32]. 

The role of HCPs in countering mis- and disinformation among their patients has also been emphasized by the Employment, Social Policy, Health, and Consumer Affairs (EPSCO) Council on the 9 December 2022, in which ministers responsible for policy on employment, social affairs, health, and consumers from EU member states and relevant European commissioners participate [33,34]. This council concluded that in the context of vaccine hesitancy, the EU member states and the European Commission should make efforts to boost HCPs’ ability to do this by providing training opportunities to HCPs. The EPSCO Council also proposed that the European Commission should set up a vaccine hesitancy expert forum with an aim to increase vaccination across the EU [35].

Some of the participants’ feedback in the survey shows that, compared to individuals, representatives of PCOs and HCPOs are more willing to vaccinate (unpublished data). This observation can probably be attributed to an increased trust in the regulatory system of these representatives. Trust in scientific and medical institutions is one of the strongest motivators for vaccine acceptance and being part of such institutions increases vaccination numbers precisely because of a lower exposure to misinformation [36,37].

During a public health emergency characterized by considerable scientific uncertainty and political leaders acting as frontline crisis managers, it often becomes a challenge to ensure that evidence-based recommendations take precedence over political interests [38]. The recent pandemic has shown how accurate communication from official sources, such as the EMA or national medicines agencies, is essential for building confidence in vaccines. A study by Roozenbeek et al. emphasizes this as well by stating that scientific information might be trusted more if people perceive its communication as transparent and open and that it might result in a reduction of people’s reliance on misinformation [39]. Fake news, especially when it relates to health, affects millions of citizens through social and digital tools. A hidden “pandemic of information”, a digital infodemic, causes enormous damage and, although digital, has a very high cost in terms of human lives in the real world [40]. Like other infodemics, the COVID-19 vaccine rumor is highly contagious, can spread exponentially around the world, and can impact overall vaccine acceptancy [41]. For this reason, the fight against misinformation must become a priority of institutions at their highest level. Listening to and analyzing its circulation through monitoring social and traditional media can be key to identifying trends in the EU and can provide a better understanding of why misinformation is spreading [42]. In this regard, it is imperative that scientific knowledge is shared through official sources and through simple but effective communication. Also, experiences at the local Italian level, which are easily extendable to the general level, show how important accurate, honest, and transparent communication is [43]. Keeping people in the dark makes them more susceptible to misinformation and conspiracy theories. People can handle the truth, even when it involves the communication of uncertainties. Half-truths expressed in the wrong way can instead prove to be a dangerous boomerang. 

As mentioned before and as described by Cavaleri et al. [44], it is first and foremost essential to listen to the needs and concerns of the population before undertaking communication activities. Only by doing this can the public’s trust in regulators be safeguarded. In order to achieve public trust, considering the context of an increased interest in and scientific scrutiny of the Agency’s work, the EMA’s communication and engagement activities needed to be adapted. Since the start of the COVID-19 pandemic and with input from the surveys conducted as part of this study, the EMA has stepped up its communication and stakeholder engagement activities in several ways. One of the initiatives was the organization of public stakeholder meetings on COVID-19 vaccines between 2020 and 2021 to attempt to reach that part of the population most at risk of misinformation. These meetings provided scientific experts a platform to address questions and concerns [19,44]. Four of these meetings were organized during the first year of the European vaccination campaign. The Agency has also made the recordings and presentations from the EMA’s public stakeholders’ meeting on COVID-19 vaccines publicly available on its website [19,45,46]. These have been updated with new content as the pandemic has progressed and as new vaccines have been approved, in line with new evidence emerging on safety and effectiveness. The EMA also keeps the COVID-19 vaccine webpages updated as per participants’ suggestions [47,48]. Secondly, visuals to accompany text explanations were incorporated. These visuals aimed to provide an explanation of relevant regulatory concepts and outcomes [19,21,49]. The majority of the participants of this study found visuals to be useful for conveying key concepts: a picture communicates better than text and this makes it an excellent way of disseminating information. Because visuals have a “long half-life”, as they are repurposed by social media channels and other platforms, it is absolutely recommended to user-test them as well to ensure effective communication and understanding by the target audience. According to a German study, visual content promotes better communication and increases confidence in vaccination [50,51]. Given participants’ requests for more audio-visual materials, the EMA also contributed to developing infographics by other institutions such as the European Council to explain in a simple and intuitive way how COVID-19 vaccines work [52,53,54]. In addition, a scale-up of the EMA’s media engagement activities took place, and pandemic safety updates were introduced and published on a regular basis for the first vaccines in clinical use [19,27,55,56].

Moreover, the COVID-19 pandemic provided the EMA with the opportunity to improve its transparency. An example of such a measure is the introduction of shorter publishing timeframes for assessment reports and assessed clinical data [19,57]. In addition, EU PCO and HCPO representatives became part of the Emergency Task Force (ETF). The ETF is an advisory support EMA body that handles regulatory activities in preparation for and during a public health emergency, such as a pandemic. By belonging to the ETF, civil society representatives could witness the transparent governance of the EMA’s crisis structures, provide input, and actively participate in discussions concerning evaluations of individual vaccines and vaccination discussions [19]. The participation of civil society representatives in the ETF is now continued as part of the EMA’s extended mandate. The extended mandate, introduced by the European Commission and applicable since 1 March 2022, aims to improve the EU’s preparedness and response to health threats and has a strong focus on transparency. Stakeholder engagement plays an important role in the various operations of the mandate and ensuring that stakeholders are given a platform to interact and discuss issues with the EMA [58].

While the regional survey revealed that information in English was not a barrier to understanding the information, the above-mentioned reasons prompt consideration of the possibility of translating more EMA documents into local EU languages to promote the wide dissemination of information to the public and to reach people with low health literacy and limited knowledge of English. The EMA already publishes translations of key documents and information of great relevance and impact in all official EU languages. This includes information on medicines that the EMA evaluates, such as overviews of authorized human medicines, medicines for which authorization has been refused, withdrawn applications, product information for authorized medicines, and referrals of medicines. Furthermore, general information on the activities and work of the Agency is translated. In addition, citizens can send inquiries to the EMA and receive a response in the same language [59]. Another possible way forward could be to translate highlights of the content into all official EU languages. 

This study needs to be interpreted in light of several limitations that may limit the generalizability of the results. First, a possible lack of representativeness due to the limited number of participants needs to be taken into account. In this regard, considering the EMA’s experience running stakeholder surveys among PCOs and HCPOs, 60 respondents in total to the EU and regional surveys can provide meaningful feedback. A second limitation could be that respondents from EU organizations have a better knowledge of the regulatory system and its outcomes, and this could represent a source of bias in the understanding of the information. On the contrary, there were more individual HCPs/patients that organizations recruited in local hospitals at the Italian regional level. Thirdly, since the surveys were mainly conducted through online recruitment and participants were volunteers, selection bias should be considered as a possible limitation. Fourth, even though an analysis of the regional Italian context revealed similarities in the understanding of key messages, it is unknown how representative this is of other EU regions. Another limitation of this study is that EU stakeholders responding to this survey were sent the Tier 1 and 2 documents in PDF format and it is not possible to establish how long they spent reading the documents.

## 5. Conclusions

Our research shows the importance of user-testing information materials during a public health crisis to guide regulators and HCPs in creating future evidence-based communication materials on vaccines. The similarities between the results at both EU and local levels reinforce the findings of this study. The proactive provision of information by authorities on COVID-19 vaccines during the public health crisis is considered important to counteract hesitancy and prebunk misinformation. The key messages and concepts tested were generally well understood, but areas of improvement were identified, including simplifying language and displaying key concepts/messages more prominently. Individual patients tend to search the internet and media more, while representatives of PCOs and HCPOs rely more on government sources and scientific publications. This indicates the importance of science communication through different channels (i.e., regulatory/scientific channels and mass media/social media), and that complementing efforts with visuals helps to convey information. Moreover, user-testing graphics revealed that the public’s preferences may differ from those of regulatory authorities. Ultimately, tailoring information materials to the needs of the public may improve trust and raise awareness of the science underpinning vaccines developed in a public health crisis.

## Figures and Tables

**Figure 1 vaccines-11-01616-f001:**
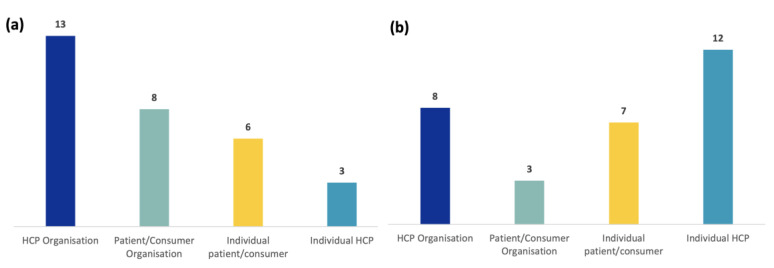
Profile of participants to EU-level survey (**a**) and Italian regional survey (**b**) by stakeholder group.

**Figure 2 vaccines-11-01616-f002:**
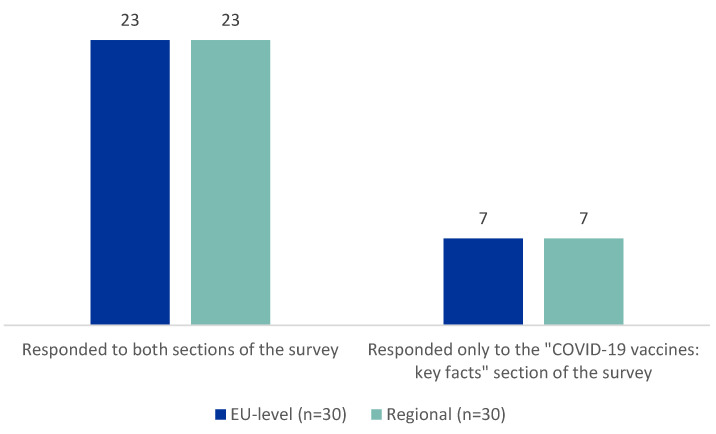
Number of participants who completed only the first section (Tier 1) or both sections (Tier 1 and Tier 2) of the survey.

**Figure 3 vaccines-11-01616-f003:**
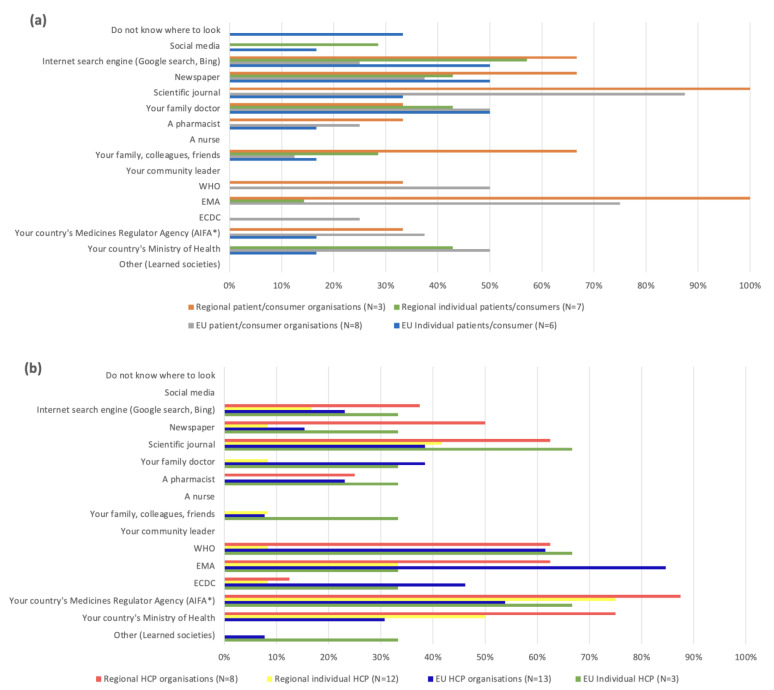
Comparison of preferred sources of COVID-19 vaccine information across stakeholder groups. (**a**). EU and regional individual patients and consumers and patient and consumer organizations. (**b**). EU and regional individual healthcare professionals and healthcare professional organizations.

**Figure 4 vaccines-11-01616-f004:**
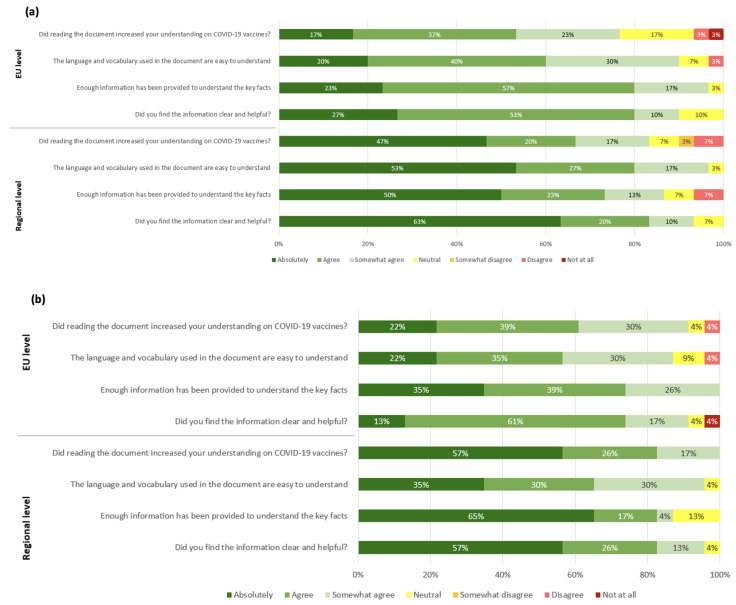
Participants’ feedback on usefulness, language, and level of detail of COVID-19 vaccine materials. (**a**) Feedback on Tier 1 “COVID-19 vaccines: basic facts” by EU-level (*n* = 30) and regional (*n* = 30) participants. (**b**) Feedback on Tier 2 “COVID-19 vaccines: development, evaluation, approval and monitoring” by EU-level (*n* = 23) and regional (*n* = 23) participants.

**Figure 5 vaccines-11-01616-f005:**
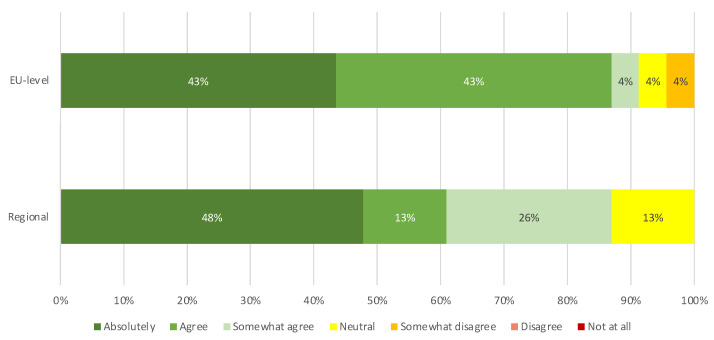
EU-level and regional participants’ feedback on whether the graphics helped to convey key concepts (Tier 2).

**Figure 6 vaccines-11-01616-f006:**
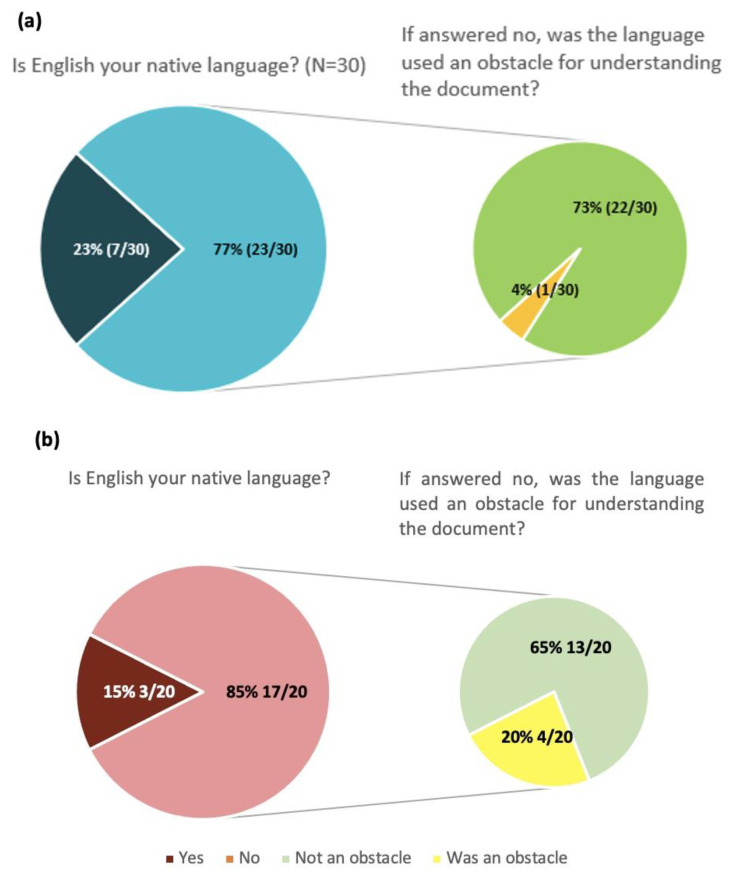
Native English language status and skills for EU-level (**a**) and Italian regional (**b**) participants. Some Italian respondents who indicated that English was their native language also responded to the question of whether the English language was an obstacle for them, which was only meant to be answered by the non-native-English-speaking participants.

**Table 1 vaccines-11-01616-t001:** The understanding of key messages of Tier 1 “COVID-19 vaccines: basic facts” and Tier 2 “COVID-19 vaccines: development, evaluation, approval and monitoring” by stakeholder groups at EU and regional levels.

	EU Level					Regional Level				
	Correct Responses by Individual Patients	Correct Responses by Individual HCPs	Correct Responses by PCOs	Correct Responses by HCP Organizations	Average Percentage across all EU Stakeholders	Correct Responses by Individual Patients	Correct Responses by Individual HCPs	Correct Responses by PCOs	Correct Responses by HCP Organizations	Average Percentage across All Regional Stakeholders
Tier 1 key messages on:										
Safety requirements ^1^	6 (100%)	2 (67%)	8 (100%)	13 (100%)	91.75%	7 (100%)	12 (100%)	3 (100%)	8 (100%)	100%
Approval ^2^	6 (100%)	3 (100%)	8 (100%)	13 (100%)	100%	7 (100%)	12 (100%)	3 (100%)	8 (100%)	100%
Safe and effective vaccines ^3^	1 (17%)	0 (0%)	3 (38%)	7 (54%)	27.3%	0 (0%)	3 (25%)	2 (67%)	4 (50%)	35.5%
Tier 2 key messages on:										
Safety monitoring ^4^	2 (100%)	2 (100%)	7 (88%)	10 (91%)	94.8%	4 (80%)	8 (73%)	N.A.	6 (86%)	79.7%
Rolling review ^5^	2 (100%)	2 (100%)	8 (100%)	11 (100%)	100%	5 (100%)	11 (100%)	N.A.	7 (100%)	100%
Development ^6^	2 (100%)	1 (50%)	6 (75%)	10 (91%)	79%	4 (80%)	7 (64%)	N.A.	7 (100%)	81.3%

Full key messages: ^1^ True or False: The safety requirements for COVID-19 vaccines are the same as for any other vaccine in the EU and will not be lowered in the context of the pandemic. ^2^ True or False: The EMA carries out a scientific evaluation of the vaccine’s safety, efficacy, and quality, before concluding on whether the vaccine can be approved. ^3^ Multiple choice: How does the EMA ensure that COVID-19 vaccines will be safe and effective? ^4^ Multiple choice: When is vaccine safety monitored? ^5^ Multiple choice: What is a rolling review? ^6^ Multiple choice: Which of the below help with the development of COVID-19 vaccines?. N.A. means not-applicable.

**Table 2 vaccines-11-01616-t002:** EU- and regional-level participants’ preferred visuals conveying the pooling of resources to fast-track medicine approval and a positive benefit–risk balance.

Visuals (Tier 2)	Level	Did You Think They Were Clear and Easy to Understand?				
		Yes	No/Don’t Know	If Answered Yes, Which Figure Do You Prefer?		
Resource pooling visuals				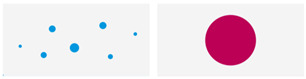 Resources COVID-19 development mobilises more extensive resources, simultaneously	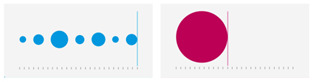 Resources COVID-19 development mobilises more extensive resources, simultaneously	
	EU	15 (65%)	8 (35%)	5 (31%)		11 (69%)
	Regional	6 (26%)	17 (74%)	0 (0%)		8 (100%)
Positive benefit–risk balance visuals				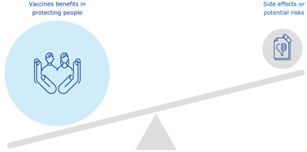 (a)	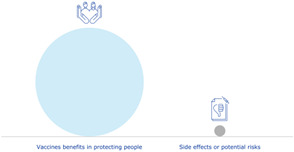 (b)	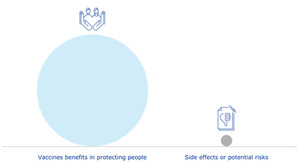 (c)
	EU	20 (87%)	3 (13%)	12 (60%)	2 (10%)	6 (30%)
	Regional	17 (74%)	6 (26%)	15 (83%)	0 (0%)	3 (17%)

**Table 3 vaccines-11-01616-t003:** Preferred safety statement by EU and regional stakeholder groups.

	EU Level (*n* = 23)				Regional Level (*n* = 23)			
	Individual Patient and Consumer Preferences	Individual HCPs’ Preferences	PCOs’ Preferences	HCPOs’ Preferences	Individual Patient and Consumer Preferences	Individual HCPs’ Preferences	PCOs’ Preferences	HCPOs’ Preferences
First safety statement proposal ^1^	2 (100%)	0 (0%)	4 (50%)	9 (82%)	2 (40%)	4 (36%)	N.A.	4 (57%)
Second safety statement proposal ^2^	0 (0%)	2 (100%)	4 (50%)	2 (18%)	3 (60%)	7 (64%)	N.A.	3 (43%)

^1^ Like any medicine, vaccines have benefits and risks, and although highly effective, no vaccine is 100% effective in preventing a disease or 100% safe in all vaccinated people. ^2^ Like any medicine, vaccines have benefits and risks, and the conditions of marketing authorization aim at maximizing benefits and minimizing risks.

**Table 4 vaccines-11-01616-t004:** Recommendations for the improvement of EMA materials after the finalization of the EU-level survey.

**Improvement Identified by Stakeholder**	**Recommendation for t** **he** **EMA**
Replace visual on positive benefit–risk balance	Replace the visual in all the EMA’s materials with the one preferred by stakeholders
Improve key messages and more friendly language	Update text and section headings
Prepare more succinct materials	* Make available on the EMA’s website the recordings and presentations of the EMA’s public stakeholder meetings, which summarize all information on COVID-19 vaccines
Produce more information on vaccine safety	Prepare a dedicated webpage on COVID-19 vaccine safety: https://www.ema.europa.eu/en/human-regulatory/overview/public-health-threats/coronavirus-disease-covid-19/covid-19-medicines/safety-covid-19-vaccines (accessed on 11 October 2023) Make available on the EMA’s website dedicated safety updates for each approved COVID-19 vaccine as needed, in addition to all information on safety routinely published for each vaccine (safety communications and summaries of scientific assessments)
Produce more friendly (audio) visual materials	In addition to (*), produce infographics, social media campaigns
Keep information updated	Keep the EMA’s webpages on COVID-19 vaccines regularly updated
Use of preferred general safety statement	Use in EMA materials the safety statement preferred by EU HCP organizations
Translations into official EU languages	Explore translating key information on COVID-19 vaccines into official EU languages

## Data Availability

Data obtained in the present study are available upon request to the corresponding author.

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
