# Peer review of "Stakeholders’ Understanding of European Medicine Agency’s COVID-19 Vaccine Information Materials in EU and Regional Contexts"

_vaccines, 2023, doi:10.3390/vaccines11101616_

Round 1

Reviewer 1 Report

In this paper, Castro et al report the results of a survey to investigate how well the EMA’s COVID vaccine information material was perceived by stakeholders in the EU, and in one Italian region (Sicily) in particular. They assessed two different documents, one about key COVID facts, and one about vaccine development and approval. They found overall that the materials were well received and understood, although some improvements were suggested for visuals, texts, and dissemination formats.

Overall, I agree with the authors’ proposal that prebunking misinformation is an indispensable strategy, achieved by providing reliable, easy-to-read information from an authoritative source. However, my main concern with this paper is that, overall, there were only 30 respondents for the EU part and 30 respondents for the Sicilian part. For an agency that represents the whole EU, with its almost 450 million inhabitants, there is no possible way that a survey of only 60 people can be representative. In addition, if these respondents are already subscribed to, or registered with a database of stakeholders willing to answer EMA surveys, then there is every likelihood that they are people who are particularly motivated by health issues, probably with a higher level of education, and/or quite pronounced opinions. This exacerbates the potential bias.

The article is well written, and the results are detailed, and the discussion appropriate. However, overall, I regret that a survey organized by the EMA could not bring together more respondents than this. I was expecting thousands of responses from a range of EU countries, which would have made the information much more valuable. I do not feel that a survey of only 60 respondents, 30 of whom come from one single region, can give a robust answer about the acceptability or value or EMA materials.

Reviewer 2 Report

This manuscript addresses the important problem of how to defeat hesitancy

to adopt vaccination against Covid-19. In the introduction they explain

why this is an urgent problem to solve. They notice that despite the unquestionable

evidence that vaccines are a reliable method to fight the spreading of infection,

 a large part of EU population remains unvaccinated.  They discuss the result

of a social experiment to establish the most efficient way to counteract hesitancy

and to prebunk misinformation, surveying patients/consumers, healthcare

professional organizations, and individual stakeholder to see how to make

the information materials well understood. They note that while professional

health care professionals rely on government sources and scientific information,

 individual patients base their decision on internet and social media that

 may be sources of misinformation. The results of this social experiment

make possible for them to find the proper language to make their arguments

comprehensible by a public, with the interesting discovery that their arguments

should be complemented using proper visuals.

I am convinced that the results of this social experiment are very important and that

this manuscript is worth of publication on the journal “vaccines”.

Author Response

Thank you very much for taking the time to review this manuscript and for highlighting its importance.

Reviewer 3 Report

This article appears to me a submission of an evaluation report of midia products developed by EMA for solving questions on COVID 19 vaccine development and revision. This type of critycal revision of midia products is interesting. 

There are a good sample in the midia and a small sample in health personal people. The authors are biased to accept the quality of the midia products they produced, as they are associated to EMA. This fact is obvious in most analysis and they provide a good report on quality of midia products, which is in fact an attempt to EMA be recognized as an adequate ethical institution for vaccine approval. The main analysis is presented without statistical analysis, without 95% CI of proportions and small samples, but with data numbers that allow reinterpretation by external authors. Their data could be also reanalyzed with different look for external authors and this is  the main  basis of my approval for publication.

I believe that the publication of an analysis when only 50% approval  of the divulging product  distributed is an interesting aspect for public health midia producers. These data could be used for constant reneval of their material.    

Reviewer 4 Report

F.B review:

Analysis of European Medicines Agency’s COVID-19 vaccine 2 information materials in EU and regional context

General impression: the manuscript deals with clarity, understanding, and impact of the European Medicines Agency’s COVID-19 vaccine 2 information materials on citizens in Europe and other local regions. The manuscript includes both a quantitative and a qualitative study. Following the results, the discussion raises some suggestions for authorities how to improve information delivery and how to increase the impact of this information to overcome obstacles delivered by social media, such as fear and misinformation. This information is important to share.

Here are some comments to the authors:

The title:

Comment # 1: the title is fine. However, it does not make the reader curious about the study i.e. what is the aim of the study or which idea the authors want to examine? Impact on willingness, clarity, relevance etc.

Abstract:

Comment # 2: the abstract does not reflect the whole study but rather the introduction and some other information. I suggest writing the objectives of the study (what was the main reason the study was done for), what were the main 4-5 finding and what the authors thing or recommend to do as following the results. This may better reflect the whole study, and make reader more curious to read the study.

Introduction:

Is fine and sufficient.  

Material and methods:

Comment # 3: the materials and methods are clear and well detailed. There was a room to compare the results where the sample was > 30. However, a smaller sample (25 participants) renders statistical analysis more complicated. Examples: paragraph line 258 and paragraph line 263.  

 Comment # 4: line175: the status of vaccination of the participant was not known. Do the authors think that this may not bias the results concerning the willingness of participants to get vaccinated when some of them had been already vaccinated? I would expect the study check the impact of EMA`s information on those who did not get vaccinated. Please refer to this point in the limitations of the study

Results:

A general comment: the results are written in a diffuse way with many details and the reader may get lost in the details. Consider concentrating on the main findings if possible. Furthermore, you have too much figures, that some can be summarized literally while some others need explanation. My note is a suggestion only to make the manuscript more confluent and less busy.

Comment # 5 paragraph line 188: the average visit time of EMA`s web site was less than one minute, what does that mean? Was that enough to understand the content? Was the tiers too long. Please deal with that in the discussion.

Comment # 6, Table 1: column 2 numbers are with apostrophe (272,977) and column 3 with dote (134.607). is that OK?

Comment # 7, line 274: what is ECDC? European CDC?

Comment # 8 line 279: what is AIFA abbreviation? It does not match Italian Medicines Agency

Comment # 9 Graph a & b: so far the results sound interesting. Please discuss the results or explain.

Comment # 10 line 360: "and agree….", agree to what? Is the sentence clear enough?

Comment # 11, paragraph line 364-367 and 367-370: both groups deals with understanding the information. However, why the results are different? Not clear to me

Comment # 12, line 389: "a stronger positive….". positive what? Attitude? Understanding? Furthermore, what do you conclude from that in the context that no statistical significance was performed? Is this information informative?

Comment # 13, line 45 (under figure 7): " The survey also asked participants to identify areas where more information is 45 needed. More information on COVID-19 vaccines safety information was requested". Did you detail the survey results somewhere? If not, please add some words.

Comment # 13, line 58: " 3.9. Exploring participants’ willingness to get vaccinated after reading the COVID-19 56 information materials", see Comment # 4.

Discussion:

A general comment: the discussion is comprehensive. However, I suggest ending each new issue with your suggestion or recommendation. Some of the paragraphs contain your recommendation in the middle of the paragraph and the reader may miss the point.

Comment # 14, line 131: "This research aimed to user-test EMA’s….", it is not clear what it aimed to do. Please see if paragraph is clear

Comment # 15, line 152: "sceptic….". did you mean skeptic?

Comment # 16, paragraph 157-169: the paragraph is too long and not clear. Consider reframing

Comment # 17, paragraph lines 179 – 190: following this paragraph and the figure 7, do you have any conclusions or recommendations in this context. The study was performed in Palermo; do you believe English language is understandable among south Italian citizens?

Comment # 18, line 241: "around the word", did you mean "around the world"?

Comment # 19: paragraph line 245: it is important but not sharp enough. What do you suggest?

Comment # 20 line 287: "input an actively ….", did you mean "input and actively…."?

Comment # 21 line 295, line 289: what is ETF

Conclusions:

Fine

Good Luck

the quality of English is fair. there are some mistakes. I wrote in my comments. needs minor revision. 

Round 2

Reviewer 1 Report

I appreciate the authors' attempts to include mention of the limitations in their paper, but I still firmly believe that 60 people cannot be considered representative of an organisation that covers a population of 450 million. 

The paper is well written. 

Author Response

We appreciate the comment from reviewer 1 and we would like to highlight that we agree with the reviewer’s remark that in view of the sample of the study, the results cannot be considered representative of the entire EU population. We believe that the current wording in the revised manuscript includes that limitation. Despite the fact that it is not possible to extract strong conclusions representing the views of the EU as a whole, the significance of our findings should not be underestimated from our perspective, and that as indicated in our revised title, our participants were stakeholders regardless.

The EU level survey received thirteen participants from European associations representing HCPOs and there were 8 respondents from patients and consumers organisations. These are in line with participation to EMA initiatives from EMA stakeholder groups. Currently, eligible organisations registered in EMA stakeholders’ database includes 38 HCPOs and 42 PCOs. Among HCPOs, there is a wide distribution of therapeutic areas, and different types of HCPs are represented (from nurses to GPs, hospital and community pharmacists and specialist doctors). Patient and consumer organisations also span different therapeutic areas, general consumer associations and special populations such as geriatric, paediatric and rare diseases.

Responders to the survey usually engage in different EMA activities and they advocate and bring the views of the individual members of the European organisations they represent. Considering that the feedback from each organisation could be presenting the collective views across their members from across EU Member States, we still consider this a relevant and meaningful input.